# Exploring the Grapevine Microbiome: Insights into the Microbial Ecosystem of Grape Berries

**DOI:** 10.3390/microorganisms13020438

**Published:** 2025-02-17

**Authors:** Daniela Minerdi, Paolo Sabbatini

**Affiliations:** 1Department of Agricultural, Forestry and Food Sciences, University of Turin, Largo Paolo Braccini 2, 10095 Grugliasco, TO, Italy; paolo.sabbatini@unito.it; 2Department of Horticulture, Michigan State University, East Lansing, MI 48824, USA; 3Interdepartmental Centre for Grapevines and Wine Sciences, University of Turin, Corso Enotria 2/C, 12051 Alba, CN, Italy

**Keywords:** plant microbiome, grapevine, grape berries, viticulture, secondary metabolites

## Abstract

Plant growth, health, and resilience to stress are intricately linked to their associated microbiomes. Grapevine, functioning as a holobiont, forms essential relationships with fungi and bacteria across both its belowground (roots) and aboveground (leaves and berries) compartments. The root microbiome exhibits a stable, site-specific structure, whereas the microbiomes of ephemeral tissues such as leaves and berries, which regenerate annually, display more stochastic assembly patterns across growing seasons. Among these, grape berries represent a critical component in viticulture due to their direct influence on wine quality and flavor complexity. Berries provide a unique ecological niche, hosting diverse microbial communities composed of yeasts, bacteria, and fungi that interact with the grapevine and its surrounding environment. These microorganisms are not only pivotal to berry development but also contribute significantly to the synthesis of secondary metabolites and fermentation processes, ultimately shaping the sensory and organoleptic properties of wine. This review consolidates current knowledge on the grapevine microbiome, with a particular emphasis on the microbial dynamics of grape berries.

## 1. Introduction

Plant growth, health, and resilience to stress are intricately linked to their associated microbiomes. This review significantly enhances the understanding of plant-associated microbiomes, particularly in grapevines, by addressing critical gaps and offering new perspectives for future research. It emphasizes the concept of the grapevine as a holobiont, integrating aboveground and belowground microbial communities to illustrate their collective role in plant health, productivity, and resilience. A key focus is the in-depth exploration of grape berry microbiomes, which are crucial for wine quality but less studied in terms of their ecological and functional contributions to stress tolerance, disease suppression, and metabolite production. This review shifts the focus from taxonomic to functional perspectives, incorporating advanced concepts such as core microbiota, keystone taxa, and microbial networks. This approach provides a framework for identifying microbial groups with potential applications in viticulture. It also addresses climate change’s impact on viticulture, proposing microbiome-based strategies to improve grapevine resilience and sustainability under environmental pressures. By advocating for the use of advanced tools like functional genomics, metabolomics, and transcriptomics, this review sets a methodological direction for unraveling microbial roles and interactions with grapevines.

The information included in this paper was extracted through a systematic search of peer-reviewed literature in databases such as PubMed, Scopus, and Web of Science. Keywords such as ’grapevine microbiome’, ’grape berries’, ’microbial diversity’, ’fungal and bacterial communities’, and ’grape and wine quality’ were used to identify relevant articles. Articles published in the last decade (2013–2023) were prioritized, with a focus on studies providing insights into microbial ecology, environmental and viticultural influences, and practical applications in viticulture and enology. Additionally, cross-references were examined to ensure comprehensive coverage of the topic. Data and conclusions from primary research papers, reviews, and meta-analyses were critically evaluated and synthesized to provide a balanced and up-to-date perspective on the grapevine microbiome.

### 1.1. Overview of Plant-Associated Microbiomes

Plants harbor highly diverse microbial communities that colonize their various belowground and aboveground tissues, forming distinct microbial habitats. The plant microbiome comprises bacteria, fungi, viruses, archaea, and protists, each playing critical roles in the functioning of these complex ecosystems. These interactions are facilitated by the ability of microorganisms to adhere to plant surfaces and infiltrate tissues, forming intricate associations. This symbiotic relationship has led to the conceptualization of the microbiome as the “second genome” of plants, representing a supplemental source of genes and functions that contribute to the plant’s adaptive potential [1]. Within this framework, the plant and its associated microbiota form a holobiont, a collective ecological unit that functions in concert [2]. Microbial interactions within the plant microbiome network can take on diverse ecological roles: mutualistic, where both partners benefit; antagonistic, where one organism benefits at the expense of another; predatory, involving direct consumption of one organism by another; commensal, where one species benefit without impacting the other; and parasitic, where one organism derives benefits while causing harm to the host.

The microbiota is acquired through vertical transmission (e.g., via seeds) or horizontal transmission from environmental reservoirs such as soil [3,4]. The soil, particularly the rhizosphere, is the most significant reservoir of plant-associated microbes. The rhizosphere is a hotspot of microbial activity and one of the most complex ecosystems, shaped by root exudates and decaying plant material that serve as nutrient sources. The chemical composition of root exudates and microbial substrate preferences significantly influence the assembly of rhizosphere microbial communities [5]. Additionally, plant-emitted volatile organic compounds (VOCs) and phytohormones contribute to microbiome modulation [6].

Beyond the rhizosphere, microbes colonize the rhizoplane (the root surface) and can enter roots as endophytes through natural gateways, such as root hairs, lateral root emergence sites, and epidermal wounds. Approximately 28% of rhizosphere microbes can colonize the root endosphere, with a smaller fraction capable of migrating to aerial plant parts [7]. Aboveground plant compartments, including leaves, stems, flowers, and fruits, also host diverse microbial communities. Microorganisms colonize these tissues via natural entry points such as stomata, trichomes, and hydathodes or through wounds [8,9]. In perennial plants like grapevines, woody structures such as trunks can act as microbial reservoirs, serving as sources for annual structures like leaves, shoots, and berries [10].

#### 1.1.1. Microbiomes in Agroecosystems

Agrosystems, including vineyards, are subject to anthropogenic influences, such as [11] tillage and dust deposition, which can introduce soil microorganisms to aerial plant surfaces [12,13]. The phyllosphere, comprising the aboveground plant surfaces, supports microbial communities that are passively transported via air, water, and soil particles [11]. Plants control the microbiota homeostasis [14] which has been shown to influence plant immunity [15]. Vegetative foliar tissues, leaves and floral parts, host endophytes as well as epiphytes. Most endophytes spread from the root via the xylem to distinct plant compartments; however, endophytes may also enter through aerial plant parts such as flowers and fruits using wounds or natural entry points such as stigma on flowers [16]. Fruits also host microbiota [17], which are often highly distinct from those found in other plant compartments such as roots [18] or bulk soil [19]. Seed microbiota may be transferred to the next plant generation [20].

#### 1.1.2. Roles of Microbiomes

Microbial interactions within plants confer numerous benefits, including enhanced stress tolerance, nutrient uptake, growth promotion [21], and pathogen resistance [22]. Beneficial microbes are involved in key processes such as nitrogen fixation, nutrient mobilization, secondary metabolite production, and induced systemic resistance (ISR) to pathogens [22]. In addition, microbial communities influence the nutritional quality and secondary metabolite profiles of plants, which are essential for crop yield and quality. In grapevines, the microbiome plays a pivotal role in plant health and productivity. The rhizosphere microbiome supports nutrient acquisition (Figure 1A) [23,24] and stress resilience (Figure 1B), while the aerial microbiota, including epiphytes and endophytes on leaves, flowers, and berries, contribute to plant immunity (Figure 1C) and quality [25,26] (Figure 1D). Grapevine berries, in particular, host highly specialized microbial communities that directly impact the synthesis of secondary metabolites (Figure 1E) and the fermentation processes (Figure 1F) critical for wine production [26]. These microbial communities are shaped by plant genotype [27], age, compartment-specific factors, environmental conditions (e.g., soil type, climate), and agricultural practices (Figure 1G) [28]. Understanding these dynamics is vital for optimizing vineyard management and enhancing wine quality.

### 1.2. Core Microbiota

The concept of the core plant microbiota refers to a stable subset of microorganisms consistently associated with a specific plant host, regardless of environmental or temporal variation [29]. These core members form the backbone of microbial interaction networks within plant-associated communities and are typically persistent and widespread across similar host populations [29]. The core microbiota often encompasses diverse microbial lifestyles, including mutualists, commensals, and pathogens, and plays a central role in shaping the structure, stability, and function of the overall plant microbiome. A key advancement in recent years has been the shift from a purely taxonomic definition of the core microbiota to a functional perspective. Instead of focusing solely on the taxonomic identity of the microorganisms, the concept of a “functional core microbiota” emphasizes the consistent traits and roles that specific microbial groups perform within the plant-associated community [30]. This functional framework has significant implications for identifying microbial groups with essential contributions to plant health and productivity. By targeting microorganisms that exhibit key functional traits, such as nutrient acquisition, stress tolerance, or pathogen suppression, researchers can uncover core microbial groups critical for plant performance, even if their taxonomic composition varies.

### 1.3. Keystone Taxa

Keystone taxa are highly connected members of the microbial network that exert disproportionate influence on the structure, stability, and function of the microbiome, regardless of their relative abundance. These taxa can act individually or as part of a guild, and their removal has been shown to cause significant disruptions to the microbial community, altering ecosystem functioning and resilience [31]. Unlike dominant taxa, which are characterized by high abundance, keystone taxa may include rare members of the community that have a unique and irreplaceable role in maintaining microbiome stability. These taxa often serve as ecological hubs, facilitating interactions between different microbial groups and ensuring the functional integrity of the microbiome [32]. Their importance extends to microbial community assembly processes, where they influence the recruitment and establishment of other microorganisms. This makes keystone taxa particularly relevant for agricultural applications, as their targeted inclusion in microbial inoculants could enhance the stability and efficacy of plant-associated microbiomes under diverse conditions.

### 1.4. Microbial Network

Microbial network analysis has emerged as a valuable tool for studying the co-occurrence patterns and interactions within plant-associated microbial communities [32]. These analytical approaches employ statistical and computational methods to construct interaction networks that visualize relationships among microbial taxa. Through these networks, researchers can identify keystone taxa and other critical nodes that regulate community structure and function [31,32]. Algorithms such as co-occurrence modeling and machine learning techniques have been increasingly applied to predict the functional roles of microbial taxa and their contributions to microbiome dynamics [33]. The integration of microbial network analysis with experimental and computational approaches provides insights into the mechanisms by which keystone taxa influence plant–microbiome interactions. For example, network-based studies have demonstrated that keystone taxa mediate essential processes such as nutrient cycling, pathogen suppression, and the modulation of plant immune responses. These findings highlight the potential of leveraging keystone taxa for agricultural applications. By incorporating keystone taxa into bioinoculants, it may be possible to engineer plant microbiomes that are more resilient to environmental stressors, more efficient in nutrient acquisition, and more effective at suppressing plant pathogens.

## 2. Grapevines and Their Importance in the World

Grapevines (*Vitis vinifera*) are among the most economically and culturally important fruit crops globally, cultivated across more than 8 million hectares and yielding an annual production exceeding 68 million metric tons. Around 70% of harvested grapes are dedicated to wine production, underscoring the species’ pivotal role in the global agricultural economy and cultural heritage. In addition to their economic significance, grape-derived products, such as juice, seed oil, and polyphenolic compounds, are recognized for their substantial health benefits. Among these, resveratrol, a polyphenol concentrated in grape skins, has been extensively studied for its cardioprotective, anti-aging, and anti-cancer properties, enhancing the societal and scientific relevance of the crop. The widespread cultivation and success of *Vitis vinifera* are largely attributed to its remarkable adaptability to diverse climatic conditions and soil types, enabling its production across a broad range of regions. However, the impacts of climate change, including rising global temperatures, altered precipitation patterns, and increased frequency of extreme weather events, present significant challenges to viticulture. These environmental pressures threaten grapevine productivity and quality, particularly in traditional wine-producing regions, where precise climate conditions are critical for maintaining the sensory and organoleptic profiles of wines [34]. Addressing these challenges necessitates the implementation of adaptive strategies in viticulture, including breeding programs focused on developing climate-resilient grapevine varieties, optimizing water and nutrient management practices, and adopting innovative technologies to mitigate the effects of climate variability. The continued success and sustainability of grapevine cultivation will depend on proactive measures to balance economic, environmental, and societal demands in a rapidly changing world.

### 2.1. Grape Berry Anatomy

The grape berry is composed of three distinct tissue types, flesh (Figure 2A), skin (Figure 2B), and seeds (Figure 2C), each contributing uniquely to the chemical composition and quality of the berry and its derived products (Figure 2). These tissues exhibit significant variation in their biochemical profiles, which directly impacts the sensory attributes and overall quality of grapes, juice, and wine. The outermost layer of the grape berry is the skin, which is covered by a hydrophobic waxy layer known as the cuticle. Unlike many other plant surfaces, the grape berry skin lacks functional stomata, resulting in limited water loss through transpiration. Consequently, the grape berry has a reduced capacity for heat dissipation via evaporative cooling and a similarly limited ability to shed excess water quickly during episodes of overhydration. This physiological trait makes grape berries particularly sensitive to environmental stressors such as heatwaves and heavy rainfall, which can compromise berry integrity and quality. Beneath the cuticle lies the hypodermis, a tightly packed layer of flattened cells that serves as a critical site for the accumulation of phenolic compounds during berry development. The flesh or mesocarp of the grape berry consists of larger, rounder cells compared to the skin, and these cells are characterized by their large central vacuoles which are the primary storage sites for sugars, acids, and water, which accumulate during berry ripening. During ripening, sucrose is enzymatically cleaved into its monosaccharide components, leading to the accumulation of glucose and fructose as the predominant sugars in mature berries. The transport of nutrients, water, and other compounds into the berry is facilitated by two primary vascular tissues: the xylem and phloem. The unique physiological and biochemical characteristics of the skin, flesh, and seeds collectively determine the quality and sensory profile of grape berries and their derived products. Understanding the complex interplay between tissue-specific processes, environmental factors, and vascular transport mechanisms is essential for optimizing grape cultivation and improving the quality of grape-derived products, particularly in the face of climate-induced challenges.

### 2.2. Grapevine Is a Holobiont

The organoleptic properties of wines, including their flavor, aroma, and texture, are influenced by a variety of factors such as vineyard location, winemaking techniques, and the microbial flora associated with both the grapes and the winery environment. These elements form the foundation of the concept of “terroir”, which integrates the biotic and abiotic characteristics of a specific region to explain the production of wines with distinctive and region-specific qualities. In recent years, the concept of “microbial terroir” has emerged as an extension of the broader notion of “viti vinicultural terroir”. This idea has gained traction due to increasing evidence that the composition of regional microbial communities, particularly bacteria and fungi, has a significant influence on wine characteristics. Such microbial communities contribute to the biochemical transformations that occur during grape maturation and fermentation, ultimately shaping the sensory attributes of the final product. They are heavily influenced by the taxonomic composition of the soil microbiome, which serves as the primary reservoir for endophytic bacteria [35,36,37] and fungi [38]. The soil microbiome, therefore, acts as a key determinant of microbial diversity and functionality across the grapevine’s various organs. The study of vineyard microbiomes has become an emerging area of research, driven by its potential to enhance grapevine adaptation to climate change, improve vineyard management practices, and mitigate the impact of pathogenic infections. Microbial diversity and richness tend to decrease along a gradient from the soil to the aerial parts of the grapevine [39,40]. Fungal diversity follows a similar trend, with the greatest richness observed in bulk soil and a sharp decline in the endorhizosphere, suggesting that root colonization by fungi is significantly restricted [41]. These patterns are particularly pronounced among epiphytic microbial communities, while endophytic populations, which can translocate within the plant via the vascular system, exhibit a more homogenized distribution across plant tissues [7]. The functional roles of the microbiota associated with aerial and underground tissues also differ markedly. Microbial communities in the rhizosphere are heavily involved in nutrient cycling, nitrogen fixation, and the production of growth-promoting compounds, while aerial microbiota contribute to plant defense mechanisms, secondary metabolite synthesis, and interactions with environmental factors such as UV light and moisture [42]. Climate, soil type, vineyard management practices, and geographical location collectively influence microbial diversity and functionality, creating a microbial signature unique to each vineyard.

#### 2.2.1. Microbiome of Underground Tissues

The most pronounced microbial differences in grapevines are observed between the external environment and the internal tissues of the roots [7,39,42,43]. Specifically, rhizospheric soil, which surrounds the roots, shows a distinct microbial composition compared to the root endosphere. Fungi belonging to the phylum *Basidiomycota*, *Chyridiomycota*, *Morteriellomycota*, *Mucoromycota*, and *Zygomycota* have also been found [12,44,45,46,47,48] (Figure 3A). For what concerns bacteria, the main phyla present in the soil, roots, and rhizosphere are *Acidobacteria*, *Actinobacteria*, *Chloroflexi*, *Firmicutes*, *Verrucomicrobia*, and *Planctomycetes* [10,49,50,51] (Figure 3A). The sharp contrast between the rhizosphere and root endosphere highlights the selective pressures exerted by the plant on its associated microbiota. The rhizosphere acts as a microbial hotspot, enriched with nutrients and signaling compounds from root exudates, which attract a wide range of microbial taxa, including beneficial, neutral, and potentially harmful organisms. However, the transition from the rhizosphere to the endosphere represents a significant ecological filter. Only a subset of microorganisms, often those capable of forming specific interactions with the plant or possessing adaptations to the endophytic lifestyle, can colonize the root interior. This selective colonization process results in a distinct microbial community within the root endosphere, where mutualistic and symbiotic relationships dominate.

#### 2.2.2. Microbiome of Aboveground Tissues

In grapevines, the aerial structures can be broadly classified into annual and perennial compartments, each playing distinct roles in plant physiology and microbial interactions. Annual structures, such as shoots, leaves, flowers, and berries, are differentiated and developed during a single vegetative cycle. In contrast, perennial structures, including the trunk, spurs, and canes, serve as reservoirs for nutrients such as sugars and nitrogen and act as long-term microbiota reservoirs. However, these woody tissues are not completely homogeneous due to the coexistence of living and dead cells, which create microenvironments that support a diverse range of microorganisms. The microbiota associated with perennial woody tissues is less variable over time compared to the microbiota of annual structures [52,53,54,55] (Figure 3B). This stability contributes to the higher microbial diversity found in woody compartments, such as the bark, compared to leaves and berries. For instance, the bacterial microbiota on epiphytic bark is reported to be more complex than that found on leaves or grape berries [45]. These perennial parts may play a crucial role as reservoirs of beneficial microorganisms, potentially replenishing annual tissues with microbes at the beginning of each vegetative cycle.

##### Phyllosphere

Among the aboveground annual structures, the phyllosphere, the surface of leaves, has been extensively studied as a key vegetative compartment. The upper and lower surfaces of leaves experience different environmental conditions, including thermal and water stresses, which influence the abundance and diversity of associated microbial communities [9,56,57,58,59,60] (Figure 3C). Despite being exposed to higher levels of ultraviolet (UV) radiation, temperature fluctuations, and moisture variability, the upper leaf surface often harbors a larger number of microorganisms compared to the lower surface [50]. These differences may be attributed to the adaptations of specific microbial taxa to withstand harsher conditions, as well as variations in nutrient availability on each leaf surface.

##### Reproductive Structures

Reproductive structures such as flowers, seeds, and berries also host unique microbial communities (Figure 3D). For example, research by Compant and colleagues [16] demonstrated that seeds are colonized by bacteria, suggesting potential vertical transmission of microbiota from parent plants to their offspring. This microbial inheritance may have implications for seedling health and early plant development, contributing to the continuity of plant-associated microbiota across generations.

##### Epiphytic Microbiome of Berries

Grape berries harbor a diverse array of microbial communities originating from the surrounding vineyard environment, many of which play critical roles in shaping the must fermentation process and ultimately influencing wine quality. These microbial populations include bacteria, fungi, and yeasts [40,53,57,60,61,62,63] (Figure 3D), which contribute to a range of biochemical transformations during fermentation, impacting the sensory attributes and chemical profile of the resulting wine. Among the aboveground parts of grapevines, the microbiota of grape berries has received the most scientific attention, primarily due to its direct role in winemaking. However, its potential influence on plant health has been less frequently studied [46]. The surface of grape berries presents an unstable and dynamic habitat, undergoing substantial changes in microbial composition and diversity as the berries progress through different stages of ripening. Factors such as berry maturity, pesticide application, and the health condition of the berries significantly influence the diversity and abundance of microbial populations associated with the grape surface [12,51,64,65,66]. For example, early stages of berry development are typically dominated by epiphytic bacteria and yeasts capable of tolerating environmental stressors, while later stages, particularly at full ripening, are characterized by increased microbial diversity and greater contributions from yeasts relevant to fermentation. Evidence suggests that there is a strong ecological linkage between soil microbiota and the microbial communities associated with grapevine aerial parts, including grape berries. Many epiphytic bacteria found on the surface of grape berries are also detected in the soil. This connection underscores the importance of soil management practices in shaping the overall microbiota of grapevines, including those directly associated with berry health and quality. The dynamic and transient nature of the berry microbiota highlights the need to better understand how these microbial communities interact with the plant and the surrounding environment. While the role of berry-associated microbes in fermentation is well established, their potential contributions to plant health, such as disease suppression or stress tolerance, remain an underexplored but promising area of research. By identifying key microbial taxa that contribute to both plant health and fermentation outcomes, it may be possible to develop strategies that optimize vineyard microbiomes for both agricultural productivity and wine quality.

## 3. Future Perspectives

Grapevines exist in close association with a diverse array of microorganisms that play significant roles in modulating their physiology throughout the plant’s life cycle. This intricate interaction between the grapevine and its associated microbial communities, collectively referred to as the grapevine holobiont, can influence the plant’s health, productivity, and resilience. Maintaining the health of the grapevine holobiont is essential for mitigating the impacts of dieback diseases and other biotic and abiotic stresses that pose a growing threat to vineyards worldwide. To address the challenges posed by grapevine dieback and related issues, it is critical to deepen our understanding of the complex relationships within the grapevine holobiont. This includes both plant–microorganism interactions and the interconnections between microbial community members themselves. The functional contributions of the microbiota, such as their roles in nutrient cycling, disease suppression, and stress tolerance, are central to the maintenance of holobiont health. However, to date, relatively few studies have utilized advanced functional approaches, such as metabolomics and transcriptomics, to unravel the specific roles of microbial communities in the grapevine holobiont. These tools hold great promise for identifying key microbial functions, metabolic pathways, and plant–microbe signaling mechanisms that underpin the health and productivity of grapevines. A comprehensive understanding of the grapevine holobiont is not only a scientific challenge but also a pressing issue for the future of the wine industry and sustainable viticulture. Leveraging microbial functions through strategies such as biocontrol, biostimulation, and biofertilization offers a pathway to reduce reliance on pesticides and chemical fertilizers, addressing key environmental and economic concerns. Biocontrol agents can suppress the activity of grapevine pathogens, biostimulants can enhance plant growth and stress tolerance, and biofertilizers can improve nutrient availability. As the wine industry grapples with the twin challenges of climate change and increasing consumer demand for environmentally friendly practices, the integration of microbiome-based strategies into vineyard management will be critical.

## Figures and Tables

**Figure 1 microorganisms-13-00438-f001:**
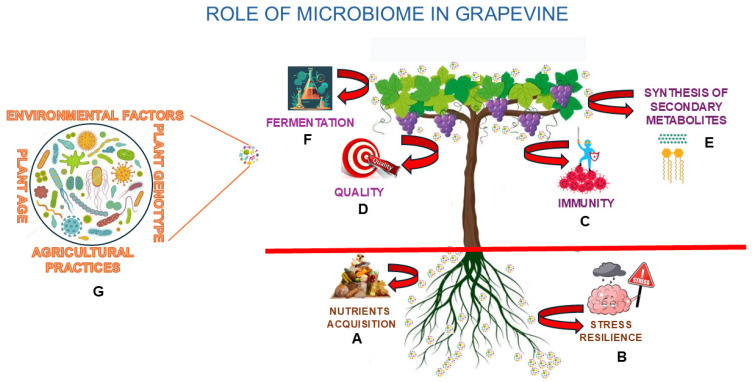
Role of microbiome in grapevine: the rhizosphere microbiome supports nutrient acquisition (**A**) and stress resilience (**B**) and the aerial microbiota contributes to plant immunity (**C**) and quality (**D**), while grapevine berries host highly specialized microbial communities that directly impact the synthesis of secondary metabolites (**E**) and the fermentation processes (**F**) critical for wine production. The microbial communities are shaped by plant genotype, age, environmental conditions, and agricultural practices (**G**).

**Figure 2 microorganisms-13-00438-f002:**
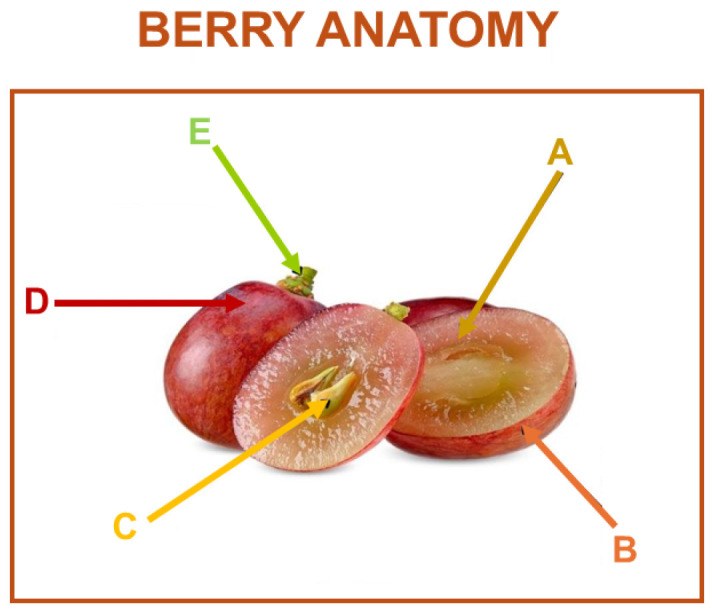
Grape pulp is the main part of the grape, consisting of water, sugars, acids, and aroma compounds. Most grapes have colorless pulp (**A**); skin holds abundant aroma compounds, precursors, tannins, and color compounds (**B**); seeds contain oils, tannins, and the embryo for potential growth (**C**); the surface is covered by a powdery waxy coating (**D**); and the stem which contains tannins connects the grape to the vine (**E**).

**Figure 3 microorganisms-13-00438-f003:**
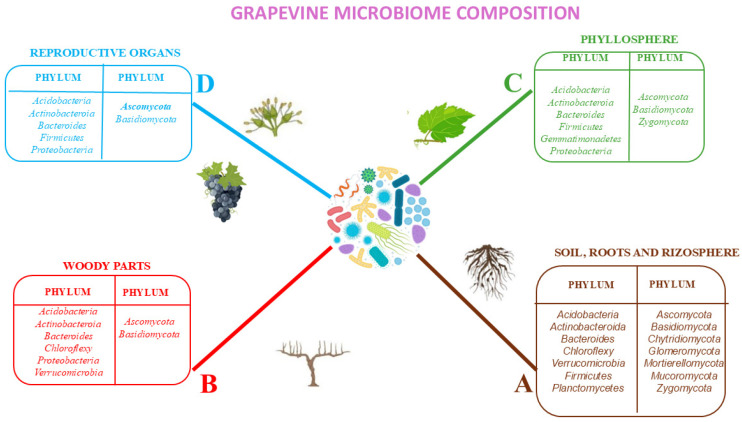
Composition of the main phyla of bacteria and fungi forming the microbiome of grapevine in underground compartments (**A**) [10,12,44,45,46,47,48,49,50,51], woody parts (**B**) [52,53,54,55], phyllosphere (**C**) [9,43,56,57,58,59], and reproductive organs (**D**) [9,40,53,56,57,58,59,60,61].

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
