# Peer review of "Exploring the Grapevine Microbiome: Insights into the Microbial Ecosystem of Grape Berries"

_microorganisms, 2025, doi:10.3390/microorganisms13020438_

Round 1

Reviewer 1 Report

Comments and Suggestions for Authors

Dear authors, the manuscript entitled "Exploring the Grapevine Microbiome: Insights into the Microbial Ecosystem of Grape Berries" present a complex review, on one of the most economically important culture and an updated knowledge for the field.

The Abstract is clear, presenting in a condensed form the main information from the manuscript.

Introduction section - I suggest the authors to add a paragraph where to explain how the manuscript is constructed, the aim and objectives of the review which will provide an insight on its sections. Also, the methodology for the extraction of information used for the review should be provided. 

The entire manuscript is well written and provide multiple information which shows its values for the filed.  

Author Response

Introduction section - I suggest the authors add a paragraph where to explain how the manuscript is constructed, the aim and objectives of the review which will provide an insight on its sections. Also, the methodology for the extraction of information used for the review should be provided

We added the paragraph in the Introduction and wrote the methodology for the extraction of information used at the end of the paragraph (lines 36-61).

Reviewer 2 Report

Comments and Suggestions for Authors

The article summarizes the recent knowledge on grapevine microbiome, particularly discussing the dynamics of microbial communities in berries.

Plant-associated microbiomes are crucial determinants of plant growth and its tolerance to environmental stresses and determine the adaptation and survival of plant hosts during extreme conditions. The growing research on the elucidation of plant-associated microbiomes has highlighted the importance of microbial communities and their dynamics in plant health, stress tolerance, and improving socio-economic traits.

The study discussing the microbiome of grapevine and its dynamics in berries, is relevant attributed to the microbial influence on fermentation, secondary metabolite synthesis affecting wine properties. Manipulation of plant microbiomes for achieving targeted outcome, is an emerging area of global research.

 I have a few suggestions for the improvement of the manuscript

Introduction: The paragraph is too long. It should be divided into small paragraphs discussing a schematic flow of information. For e.g. Overview of plant-associated microbiomes, classification, functions in plants, providing examples of plant microbiome studies, and its plant/environmental impact, etc.

Line: This review aims to delve…..should start from another paragraph.

A more coherent and systematic organization of the review is needed.

 How the present literature survey add to the growing information on plant-associated microbiomes and contribute to future research in grapevine? Discuss.

 Figure 1. Is not clear, font is not appropriate. Kindly provide a higher resolution image, 300 dpi or more for clarity. In addition, 1A, stress resilience, not clearly represented, nutrient acquisition etc. The figure needs to be redrawn providing a clear description, Fig 1A, figure 1B and 1 C should be related to each other.

 In the paper, paragraphs are too long and difficult to comprehend. Please reorganize into shorter sections for readability.

Figure 2. a higher resolution image is needed. Moreover, make sure the contents of the figure are clearly and briefly explained in the caption.

Moderate English revision for spelling, grammer etc. is required. For e.g. 2.5. Ephyphitic microbiome of berries….change to Epiphytic microbiomes…..

Recent references should be included.

Author Response

Introduction: The paragraph is too long. It should be divided into small paragraphs discussing a schematic flow of information. For e.g. Overview of plant-associated microbiomes, classification, functions in plants, providing examples of plant microbiome studies, and its plant/environmental impact, etc.

Introduction has been divided into small paragraphs: 1.1 Overview of plant-associated microbiomes, 1.1.2 Microbiomes in agroecosystems, 1.1.3 Roles of microbiomes, 1.2C ore microbiota, 1.3 Keystone taxa, 1.4 Microbial network and a schematic flow of information has been added.

Line: This review aims to delve…..should start from another paragraph.

We followed the referee’s suggestion and moved the sentence.

A more coherent and systematic organization of the review is needed.

The paper has been re-organized.

 How the present literature survey add to the growing information on plant-associated microbiomes and contribute to future research in grapevine? Discuss.

Discussion has been added.

 Figure 1. Is not clear, font is not appropriate. Kindly provide a higher resolution image, 300 dpi or more for clarity. In addition, 1A, stress resilience, not clearly represented, nutrient acquisition etc. The figure needs to be redrawn providing a clear description, Fig 1A, figure 1B and 1 C should be related to each other.

The figure has been redrawn changing the font and the description.

 In the paper, paragraphs are too long and difficult to comprehend. Please reorganize into shorter sections for readability.

Paragraphs have been reorganized in shorter sections.

Figure 2. a higher resolution image is needed. Moreover, make sure the contents of the figure are clearly and briefly explained in the caption.

Image 2 has been improved and the content has been clearly and briefly explained in the caption.

Moderate English revision for spelling, grammer etc. is required. For e.g. 2.5. Ephyphitic microbiome of berries….change to Epiphytic microbiomes…..

English has been checked.

Recent references should be included.

Recent references have been included from reference 47 to reference 75.

Round 2

Reviewer 2 Report

Comments and Suggestions for Authors

The manuscript has been considerably improved for a better expression of the theme and clarity. It can be considered for publication.